# Molecularly Imprinted Polymers Based on Chitosan for 2,4-Dichlorophenoxyacetic Acid Removal

**DOI:** 10.3390/ijms232113192

**Published:** 2022-10-29

**Authors:** Ilaria Silvestro, Marta Fernández-García, Clarissa Ciarlantini, Iolanda Francolini, Annamaria Girelli, Antonella Piozzi

**Affiliations:** 1Department of Chemistry, Sapienza University of Rome, Piazzale A. Moro, 5, 00185 Rome, Italy; 2Institute of Polymer Science and Technology (ICTP-CSIC), Juan de la Cierva 3, 28006 Madrid, Spain; 3Interdisciplinary Platform for Sustainable Plastics towards a Circular Economy-Spanish National Research Council (SusPlast-CSIC), 28006 Madrid, Spain

**Keywords:** chitosan, chitosan functionalization, molecularly imprinted polymers, 2,4-dichlorophenoxyacetic acid, pollutant removal

## Abstract

The development of low-cost and eco-friendly materials for the removal of pollutants from water is one of the main modern challenges. For this purpose, molecularly imprinted polymers were prepared under optimized conditions starting from chitosan (CS), chemically or ionically modified with glycidyl methacrylate (GMA) or itaconic acid (ITA), respectively. 2,4-Dichlorophenoxyacetic acid (2,4-D) was used as a template, obtaining the CS_GMA and CS_ITA series. The influence of the template concentration on the MIPs’ (molecularly imprinted polymers) morphology, thermal behaviour and swelling ability, as well as on the 2,4-D removal capacity, were analyzed. The amount of the template used for the imprinting, together with the different permeability of the matrices, were the key factors driving the analyte uptake process. Despite the good performance shown by the non-imprinted CS_GMA sample, the best results were obtained when CS_GMA was imprinted with the highest amount (5%) of template (CS_GMA_5). This system was also more efficient when consecutive adsorption experiments were carried out. In addition, CS_GMA_5 had a desorption efficiency of 90–100% when a low pesticide concentration was used. These findings suggest that the presence of imprinted cavities could be useful in improving the performance of sorbent materials making CS_GMA_5 a possible candidate for 2,4-D removal.

## 1. Introduction

Nowadays, the contamination of living aquatic species is a problem of great concern. Recently great efforts have been made to develop effective strategies to remove contaminants from water. 2,4-Dichlorophenoxyacetic acid (2,4-D) is a worldwide-used herbicide recognized as being one of the most ubiquitous phenoxy acids present in the aquatic environment [1]. Even if slightly soluble in soil, the danger of 2,4-D comes from being strongly soluble in water (900 mgL^−1^) [2]. Inhalation and adsorption through the skin are all possible ways of intake, which could cause liver, kidney, muscle and brain tissue damage [3]. So far, to effectively remove organic pollutants from wastewater, several analytical techniques such as precipitation, oxidation, bio- or photocatalytic-degradation and physical adsorption have been developed [4,5]. However, in some cases, the use of these techniques is limiting in terms of accuracy due to the low analyte concentrations and poor selectivity towards a particular pollutant present in complex environmental samples. Therefore, novel highly selective and reproducible systems not sensitive to the composition of the matrix and with features of sustainability need to be developed. Among the most employed approaches for pollutant removal, the use of sorbent materials is very promising. Several types of adsorbents have been developed such as synthetic materials (polymeric resins, zeolites or aluminosilicates), treated and non-treated natural materials (activated carbons silica gel, sawdust and wood), as well as agricultural waste or industrial by-products and biosorbents [6]. Obviously, choosing low-cost and environmentally friendly systems can be extremely beneficial in reducing both process costs and environmental impact. For instance, cellulosic waste biomass has shown a good adsorption ability with performances strictly related to biomass composition and its functionalization. Indeed, the high pesticide adsorption obtained by Saeed et al. using cantaloupe seed shell powder was attributed to the variety of functional groups present in the matrix [7]. Instead, Xia et al. achieved excellent results in the removal of different dyes from wastewater with the use of an innovative composite adsorbent constituted by Juncus effusus fibres modified with a natural polymer, chitosan (CS) [8]. Furthermore, hybrid materials constituted of biopolymers and inorganic nanomaterials have also been widely studied. For instance, the introduction of graphene oxide in alginate porous hydrogels remarkably improved ciprofloxacin adsorption [9], while thin layers of biopolymers on manganese ferrite used to have a better Pb(II) removal from aqueous solutions [10]. Chitosan (CS) is a natural biopolymer belonging to the group of green polymers. Its benefits derive from being biodegradable, non-toxic, biocompatible, and bioactive [11,12]. These properties, combined with the advantage of having many functional groups on its backbone (-OH and NH2), have made CS particularly interesting for many applications such as tissue engineering [6,7,8,13,14,15], bioremediation [16,17,18], agriculture [19], cosmetics [20] and as sorbent materials [21,22,23] for a wide range of contaminants. CS possesses a suitable reactivity to be modified by chemical or physical processes with the aim of widening its use. In fact, CS is also well known for its weak mechanical properties and low dimensional stability in acidic media [24]. Moreover, functional groups can promote the adsorption process benefiting pollutant removal. Nevertheless, much effort is still needed to increase the recognition ability, selectivity, and efficiency of the sorbent materials. Among the current strategies to deal with this issue, there is the use of molecularly imprinted polymers (MIPs). MIPs are the result of a synthetic process based on the arrangements of a monomer and a crosslinker around a target molecule used as a template [25]. The combination of these components produces a synthetic polymer with specific cavities able to recognize the target analyte. After the polymerization phase, the template is removed, thus obtaining an imprinted system. The molecular memory introduced into the polymer permits the selective rebinding of the target molecule. Compared to classical sorbent materials, MIPS are very interesting materials since, in addition to having molecular recognition sites, they are thermal and chemical stable systems, relatively cheap and easy to tailor to new analytical targets. Among the different approaches for MIPs preparation, non-covalent imprinting is preferred for its simplicity and versatility [26]. Therefore, hydrogen bonding, van der Waals forces, π-π and hydrophobic interactions, and electrostatic forces are exploited to form a stable template-monomer complex. A wide range of materials has been employed to synthesize MIPs [27]. The most famous functional compounds are the acrylate molecules since they are easy to polymerize. In this framework, great efforts have been made to increase the applicability of natural polymers such as CS as a selective sorbent to gradually replace non-biodegradable materials. Recently, due to the outstanding characteristics of chitosan, MIPs based on this polymer have been widely used for several applications including drug delivery [28,29], sorbent phase for solid phase extraction [30,31], pollutant removal and separation strategies [32,33]. As far as 2,4-D removal is concerned, different kinds of polymers have been tested including CS [34,35,36,37,38,39,40]. To our knowledge, no studies have reported the use of chitosan-based MIPs for this purpose so far. In contrast, the use of chitosan played a marginal role in the design of MIPs [41,42]. The aim of this work was to develop CS-based MIPs and study their possible employment for the removal of 2,4-D from water. For this purpose, CS was grafted with acrylate functional groups to make it suitable for bulk polymerization by using two strategies: covalent modification with glycidyl methacrylate (GMA) and ionic modification through its interaction with itaconic acid (ITA). The modified chitosan was used for the preparation of two different hydrogels with a mixture of monomers including ITA, GMA and dimethyl itaconate (DMI), a derivate of the itaconic acid. Except for GMA, which is a health and environmental hazard, to obtain eco-friendly systems, molecules with low environmental impact were used for MIPs production. In fact, itaconic acid was primarily produced through the carbohydrate fermentation with the fungus Aspergillus Terreus and represents today, along with its ester derivatives such as dimethyl itaconate, a valid and eco-friendly alternative to acrylic and methacrylic acid as a building block for polymerization processes [43,44,45]. 2,4-D was used as a template during the crosslinking phase to form MIPs and the effects of the imprinting obtained by using different amounts of the template (1.3–9.5 wt%) on physical–chemical properties and pesticide removal capacity of the hydrogels were studied. In addition, non-imprinted polymers (NIPs) were also prepared by employing the same synthetic procedure but without the template.

## 2. Results and Discussions

In this research, to obtain an efficient and reversible sorbent phase for the adsorption of 2,4-D, a widely used herbicide, chitosan-based MIPs were synthesized. First of all, CS was functionalized using a covalent bond with GMA and ionic interactions with ITA, thus obtaining the samples named CS:GMA and CS:ITA, respectively. Then, on the basis of the different reactivity of the modified-chitosan derivatives, CS:GMA and CS:ITA were polymerized with a mixture of monomers such as ITA, DMI and GMA (see Materials and Methods Section 3.2) obtaining two different systems. 2,4-D was used as a template in different amounts to obtain complementary binding cavities with suitable shapes and positions of the functional groups to favour herbicide uptake. The acronyms of the prepared MIPs are reported in Table 1. The properties of such systems were compared with those of NIPs (non-imprinted polymers) to evaluate the efficacy of the imprinted cavities towards 2,4-D molecule adsorption.

### 2.1. Determination of Acrylation Degree of CS Functionalized with GMA

The structural characterization of modified chitosan with glycidyl methacrylate was carried out by ^1^H-NMR spectroscopy. In Figure 1A,B, the spectra of pristine CS and a CS:GMA sample are displayed as an example. Typical signals of chitosan appeared at 5.07 ppm of H-1 of GlcN and GlcNAc part (GlcN = D-glucosamine unit and GlcNAc = N-acetyl-D-glucosamine unit). The broad group of signals in the range 3.5–4.3 ppm represented the H-3–H-6 hydrogens bonded to the non-anomeric carbons in the glucopyranose ring, while at 3.3 ppm the resonance of H-2 of GlcN residues was found. Finally, the signal at 2.4 ppm was attributed to the CH_3_ residue of the acetylated units, and the signal at 4.7 ppm to D_2_O [46]. According to the literature, in a protic solvent such as water, the reaction between polysaccharides and GMA involves the epoxide ring opening, and not the transesterification reaction, which exploits hydroxyl and carboxyl functional groups. Therefore, because of the epoxide ring opening mechanism, the formation of two products (methacryloyl-1-glyceryl ester and 3- methacryloyl-2-glyceryl ester) is possible given the two different reactive positions of the epoxide ring [47]. In Figure 1C, the scheme of the reaction mechanism is reported.

After the reaction with GMA, two characteristic peaks related to the hydrogen atoms of the vinyl group were displayed at 5.9 and 6.3 ppm as well as the signal of the methyl group of GMA at 2.10 ppm [48]. Signals of the glycerol spacer were in the range of 3.5–4.3 ppm [49]. After doing a baseline correction, the area of specific signals was recorded (see Materials and Methods) to determine the acrylation degree (DA) which was found to be around 20–22%.

### 2.2. Characterization of the Prepared CS_GMA and CS_ITA MIPs and NIPs

#### 2.2.1. Fourier Transform Infrared (FT-IR) Spectroscopy

In Figure 2, the FT-IR spectra of CS_GMA and CS_ITA NIPs and some MIPs, selected as examples, are reported, and compared with the pristine CS spectrum.

However, in the spectra of all systems, typical signals of CS were present. In particular, in the range 3500 and 3000 cm^−1^ the absorption band related to the –OH and -NH stretching was visible. The absorptions at 2920–2875 cm^−1^ were due to the C-H stretching while those at 1650 cm^−1^ and 1560 cm^−1^ were attributed to the C=O stretching of acetylate groups (amide I) and to the N-H bending of primary amine, respectively. Finally, absorptions in the range 1150–1000 cm^−1^ were related to the pyranose ring while the C-O-C and C-O-H stretching were present at 895 cm^−1^.

No significant difference was observed between the CS_GMA_X and CS_ITA_X systems, as they were prepared from the same components. Moreover, the typical peaks of the saccharide structure of chitosan hide most of the characteristic bands of the monomers used. Nevertheless, some variation from the pristine CS spectrum is still identifiable in all prepared samples. In fact, the peak at 1710–1712 cm^−1^ is due to carboxylic groups present in ITA, GMA and DMI that were used to form the network. Furthermore, the enlargement of the absorption band in the range 3600 and 2950 cm^−1^ can be attributed to the presence of the OH groups of carboxylic groups of the monomers.

#### 2.2.2. Field Emission Scanning Electron Microscopy (FE-SEM) and Energy Dispersive X-ray Spectroscopy (EDX) Analysis

The surface morphologies of MIPs and NIPs were studied by the FE-SEM/EDX technique (after the washing phase) to detect possible changes caused by the imprinting process. As an example, surface and bulk micrographs of CS_GMA and CS_GMA_5 are shown in Figure 3. CS_GMA is characterized by a quite homogeneous and compact structure as evidenced by the analysis of the bulk (Figure 3C). A quite similar structure was also evidenced by CS_GMA_1.3 and 2.5. However, the morphology was modified after the imprinting process performed with a high amount of the template (5%). In fact, a slightly rougher surface and few pores were detected for CS_GMA_5 (Figure 3B). A slightly less dense and fibrous structure was also observed when the bulk was investigated (Figure 3D). Probably, in this sample, the removal of the greater amount of the template caused relevant morphological changes.

As for the CS_ITA samples, the morphology of the systems seemed not to be strongly influenced by the imprinting process. In Figure 4, the micrographs of the CS_ITA NIP (Figure 4A,C, surface and bulk, respectively) and CS_ITA_5 MIP (Figure 4B,D, surface and bulk, respectively) are displayed as an example. As it can be noted, a homogeneous structure characterized both the bulk and surface of the samples. Moreover, CS_ITA samples showed a quite peculiar structure made by agglomerated particles. This can be explained considering the high number of ionic interactions in the CS_ITA sample can favour the formation of connected spherical particles during the polymerization phase.

To check for any other traces of the unremoved template, EDX measurements were carried out both on the surface and in the bulk of the samples. In Figure 5, the EDX spectra of some MIPs and NIPs are shown as an example, while in Table 2 the ratio of atomic % of chlorine with respect to atomic % of carbon is reported. Here, the presence of chloride (component of 2,4-D molecule) was used as a possible indication of any unextracted template in the samples. The amount of chlorine present on the surface of the imprinted systems highlighted traces of pesticide remaining in the matrices. By analyzing the ratio reported in Table 2, it was noticeable that this value rose with the increase of the amount of the template used for imprinting, thus demonstrating a greater difficulty in the removal of the pesticide. This was more evident for the sample CS_ITA_X where a higher chlorine content was found. Interestingly, no residual template was found in the bulk of the CS_GMA_X samples. This suggested that in the case of CS_GMA_X samples, it was easier to remove the template. It can be supposed that the cleaning solvent was able to penetrate the matrix favouring the migration of the template through the bulk. However, traces of the template remained trapped in the underlying surface layers suggesting the necessity of further washing steps.

On the contrary (except for CS_ITA_5 which probably had a more permeable structure), CS_ITA_X samples conserved a discrete amount of the template also in the bulk of the material. In all the samples, in addition to chromium, which was used for sputtering, sulfur was detected. This is probably due to the presence of both the residual DMSO and the initiator used for preparing the MIPs.

#### 2.2.3. Thermogravimetric Analysis

Thermal analysis allowed the degradation temperature (T_d_) of the prepared samples to be determined. The values are reported in Table 2. Specifically, after the washing phase, the samples were dried and tested. From the data, it was noted that the T_d_ value of CS_GMA was slightly lower than those of the imprinted samples CS_GMA_X. This behaviour could be explained by considering that CS_GMA, before imprinting, offered a significant amount of free NH_2_ (groups belonging to CS and not involved in the reaction with GMA), able to ionically interact with the carboxylic groups of 2,4-D pesticide in the subsequent reaction phase (Figure 6).

Therefore, the pesticide molecules, which remained bound to the samples, were able to contribute to the CS stabilization (slightly increase in the T_d_ value) thanks to the interactions among the aromatic portions of the pesticide molecules. In addition, the polymerization in presence of 2,4-D could have led to the formation of a more interpenetrated network (higher T_d_) that reduced the free volume available for the polymeric chains. Interestingly, the T_d_ value of CS_GMA was also lower than those of the CS_ITA systems. Furthermore, in this case, the interactions among the chains could explain the behaviour of the samples. In fact, in CS_GMA, the use of glycidyl methacrylate, which has a discreet steric hindrance, could have led to the formation of a larger mesh network with fewer interactions among the chains. While in the case of CS_ITA, functionalized with a smaller molecule, a better interpenetration between the components could occur with the formation of more chain interactions (higher T_d_). When the 2,4-D residual molecules were present in CS_ITA_X samples, a general decrease in the degradation temperature was observed, suggesting that the template molecules disturbed the structure of CS_ITA. This is also more plausible since a discrete amount of template was found also in the bulk of the systems. In Figure 7, the TGA and derivative (DTGA) curves of selected samples are shown, as an example. By observing the thermograms, it is noticeable that for all samples three stages of degradation are present. In the range of 25–170 °C, the first degradation was related to water evaporation but also to an initial deacetylation process and to the polymer degradation with the release of gases as water, NH_3_ and CO_2_. Following, the second step was due to the decomposition of the crosslinked polysaccharide matrix. Finally, the shoulder situated at a higher temperature was attributed to the degradation of methacrylic chains.

#### 2.2.4. Water Uptake

To determine the swelling behaviour of hydrogels, an aqueous solution at pH 5 was chosen as the swelling medium since subsequent pesticide sorption was carried out in these conditions. In Figure 8, the water uptake kinetics for both CS_GMA and CS_ITA, MIPs and NIPs, are reported. Moreover, in Table 2 the values of maximum swelling are provided. In general, the achieved swelling percentages suggested that the systems behave like super hydrogels, showing a sorption capacity up to several hundred or thousands of times their dry weight [50,51]. Such behaviour was particularly evident for the CS_GMA that absorbed a high amount of water, confirming that it was characterized by a less dense structure. This was also true for CS_GMA_1.3 which reached values comparable with CS_GMA. However, such a high swelling made these samples more difficult to handle. As a result of the imprinting process with a higher template amount, a decrease in water uptake was observed for the CS_GMA_2.5 and CS_GMA_5 samples. As previously mentioned, the increasing amount of the template used in the polymerization phase could lead to the formation of a less permeable structure with a reduction of free volume for the polymeric chains. In addition, the 2,4-D residual presence on the surfaces can contribute to the enhancement of the hydrophobicity of the systems. In general, CS_ITA and CS_ITA_X samples showed a quite low water uptake if compared with CS_GMA and CS_GMA_X, probably due to the denser network structure.

Moreover, no clear correlation between the presence of residual 2,4-D and the swelling ability of the samples was observed. Indeed, although some of the pesticide molecules remained in the matrix, the CS_ITA_5 and CS_ITA_9.5 samples showed a swelling ability higher than that of CS_ITA_2.5 (see Figure 8B). It was hypothesized that in such a compact structure, the greater amount of pesticide used in the imprinting process left more free spaces accessible to water molecules, after the pesticide removal. In the case of the CS_ITA_2.5 sample, probably the presence of the residual pesticide on the surfaces and in the bulk made the system more hydrophobic than CS_ITA NIP.

### 2.3. Pesticide Adsorption Kinetics and Desorption Test

Adsorption kinetics were performed by using a specific pesticide concentration (100 ppm) and a pH = 5. This latter condition was chosen because a pH lower than the CS pK_a_ (equal to 6.5 [52]) could ensure more interactions between the matrix and the pesticide with possible benefit for the adsorption of the pesticide itself [40]. In Figure 9A,B, the adsorption kinetics of the pesticide are reported. As can be seen, with the contact time increasing, the adsorption rate of the pesticide rapidly increased in the first 20 min and then decreased reaching the equilibrium at different times. In particular, in the first few minutes, the adsorbed pesticide amount augmented thanks to a large number of binding sites available on the surface of the matrices. Then, the adsorption rate slowed down because, since most surface sites were occupied by 2,4-D, the pesticide molecules were forced to migrate towards the imprinted sites present in the pores. This behaviour was more evident for the CS_GMA series, where the slower penetration of 2,4-D was caused by the greater difficulty of the pesticide molecules to reach the innermost binding sites, given the higher porosity of these matrices (see swelling data). The fair amount of unremoved pesticide as well as the presence of a dense structure in the CS_ITA matrices were the responsible factors for the faster surface saturation of these series. To better understand the adsorption mechanism influencing the pesticide uptake rate, pseudo-first-order and pseudo-second-order models were fitted to the experimental data. Through the analysis of R^2^ values, it was found that the adsorption process obeyed the pseudo-second-order kinetic model (Figure 9C,D). Moreover, the q_e,cal_ (theoretical adsorption capacity at equilibrium) values were in good agreement with the experimental data (q_e_, adsorption capacity at equilibrium) shown in Table 3. This behaviour suggests the existence of a relationship between the adsorption capacity and the active sites of the absorbents. Therefore, the chemical nature of the matrices drove the adsorption process as a result of the electrostatic interactions between the analyte and adsorbents (see Figure 9), as already demonstrated in previous studies [8,38,53,54].

In addition, the adsorption occurred first on the external surface and then through the pores of the absorbent material. This was particularly true for NIPs where the imprinted cavities were not present and the pesticide adsorption was predominantly physical but strongly affected by the matrix swellability. Indeed, the non-imprinted system CS_GMA and the CS_GMA_1.3 sample, where low imprinting was probably obtained, adsorbed at equilibrium the higher amount of 2,4-D (q_e_ = 4.2 mg/g and 4.0 mg/g, respectively).

This behaviour can be related to the higher swelling ability of these two systems. However, the CS_GMA_2.5 sample, which had a water uptake ability higher than CS_GMA_5, was able to absorb a low pesticide amount (2.2 mg/g vs. 3.4 mg/g, respectively). This result may be due to a greater number of cavities present in the CS_GMA_5 sample as well as the unremoved pesticide. Indeed, both factors could have favoured the interactions of the new pesticide molecules with the matrix.

Regarding the CS_ITA MIPs and NIP, the absorbed amount of 2,4-D was in a range of 2.3–3 mg/g, with a slight decrease in the adsorption capacity of the systems as the amount of template used for the imprinting increased. In this case, probably most of the cavities of the samples were still occupied by the unremoved template (see Table 2). To verify how much of the absorbed pesticide could have been desorbed from the matrices, the desorption efficiency (DE) was calculated. Among the CS_GMA_X samples, the highest value of DE was reached by CS_GMA_2.5, probably because this sample absorbed the lowest amount of pesticide or due to the presence of the residual template that provoked more superficial adsorption of the new pesticide thus making it easier to be removed. A peculiar behaviour was shown by the CS_GMA_1.3 sample that reached only 45% of DE. Since this system showed high hydrophilicity in the water uptake experiments, the pesticide may have been absorbed more deeply into the matrix in the specific cavities formed by the imprinting process, making the pesticide elution more difficult. In general, a DE percentage ≥ 64% was obtained for most of the samples underlining reasonably good reversibility of the matrices.

Although the best 2,4-D loading results were achieved by NIPs, it is not possible to conclude that MIPs were inefficient. In fact, regeneration and good reusability are fundamental properties that can reduce the overall costs of wastewater treatment [51]. For this reason, the most promising sample was chosen for further experiments where 3 consecutive cycles of 2,4-D sorption/desorption were performed. The AE (%) values were collected for each cycle. For this analysis, CS_ITA samples were excluded for their poor performance. Instead, CS_GMA_5 was selected because for this sample it could be possible to verify a positive effect given by the imprinting process. In fact, CS_GMA_5 had a similar performance to that of CS_GMA. While for this latter sample the good AE (%) was mainly due to its high swelling ability, for CS_GMA_5 was due to the presence of specific cavities. The same samples were also tested by using a pesticide solution of 10 and 50 ppm. Figure 10A shows the trend of AE (%) for 3 cycles performed at 100 ppm of 2,4-D solution. It was evidenced that the AE values decreased by about 15 percentage points after the first cycle for both samples suggesting a partial occupation of the membranes due to the previous adsorption step. Interestingly, moving to the third cycle, the AE (%) of the CS_GMA_5 sample remained unvaried while CS_GMA underwent a further decrease. Moreover, the DE of CS_GMA_5 (calculated as the ratio between the amount of desorbed pesticide and the amount absorbed plus that not removed at each cycle), was higher than the one of CS_GMA (see Figure 10B). This finding suggests that, for CS_GMA_5, the presence of the molecular memory because of the imprinting process could ensure reversibility [46,47,55,56]. Instead, for CS_GMA the faster saturation process caused a reduction in pesticide re-loading. By employing different concentrations of 2,4-D, it was evidenced that the q_e_ increased when the pesticide concentration rose from 10 to 100 ppm. CS_GMA_5 confirmed a lower value of q_e_ for all the concentrations tested if compared with CS_GMA (Figure 10C). As far as the desorption ability is concerned, the DE (%) of CS_GMA was lower than the one of CS_GMA_5 when concentrations of 10 and 50 ppm were used (Figure 10D). Specifically, the latter sample had a DE (%) between 90–113 % (values higher than 100% are accepted considering the possible contribution of a small amount of the template entrapped in the matrix). Even if a lower amount of pesticide has been absorbed, this behaviour indicates that CS_GMA_5 was more inclined to pesticide desorption. Once again, the performance showed by CS_GMA_5 can be traced to the presence of imprinted cavities that improved the reversibility of the pesticide uptake/release phase. When the pesticide concentration was increased to 100 ppm, a desorption efficiency of approximately 65% was found for both the tested samples. Here, a portion of the absorbed pesticide was probably distributed also out of the imprinted zones causing a decrease of DE (%) for CS_GMA_5. Given the variety of sorbent materials in use, it is difficult to compare our results with those of previous studies. However, some investigations have reported good results regarding the removal of 2,4-D. For instance, Zhang et al. reached a removal efficiency of 50% with good reusability by using laccase immobilized on chitosan [57]. While, in the work of El Harmoudi et al., kinetic studies showed 67 and 90% of adsorption, respectively, when chitin and chitosan were tested for 2,4-D removal [37].

A maximum adsorption capacity of 2,4-D equal to 16.92 mg/g was found by Han et al. by using a hybrid composite of magnetite/chitosan [58]. Nonetheless, the results obtained in our work are promising as they evidenced that the synthesis of MIPs from acrylate-modified chitosan can be a viable and effective route for designing eco-friendly systems for 2,4-D removal.

## 3. Materials end Methods

### 3.1. Functionalization of Chitosan (CS) with Glycidyl Methacrylate (GMA) and Itaconic Acid (ITA)

By following a literature procedure with some modifications [48], chitosan from shrimp shells, with a deacetylation degree ≥75% (Sigma Aldrich), was dissolved in 2% *v*/*v* acetic acid to obtain solutions at 2% *w*/*v*. After 24 h of stirring, the solution was degassed under an argon atmosphere for 20 min. A small amount of hydroquinone (H) was added before degasification as a stabilizer. At the end, a specific amount of GMA (97%, containing 100 ppm monomethyl ether hydroquinone as an inhibitor, Sigma Aldrich) was added drop by drop in the CS solution to obtain a 1:2 CS_repetitive unit_/GMA mole ratio. The reaction was carried out, under an argon atmosphere, at 70 °C for 24 h. Then, the polymer solution was dialyzed for 3 days to remove unreacted reagents and then was frozen and lyophilized. The reaction was conducted using 1 L of chitosan solution in a double-jacket reactor with a volume of 2 L. The difficult mechanical stirring of the system led to the formation of an inhomogeneous solution consisting of crosslinked macroscopic GMA particles and a polymer solution of modified chitosan. The solution, after being carefully filtered and separated from the GMA particles, was used for the preparation of MIPs. The sample was named CS:GMA. As for chitosan samples modified with itaconic acid, the formation of the salt between the two components was performed by dissolving chitosan (3% *w*/*v*) in distillate water containing itaconic acid, obtaining a 1:4 CS_repetitive unit_:ITA mole ratio. The solution was left stirring until complete polymer dissolution and used without further treatment. The sample was named CS:ITA.

### 3.2. Preparation of MIPs and NIPs Hydrogels with Modified Chitosan

The CS:GMA sample was used to prepare MIPs and NIPs hydrogels. The samples were named CS_GMA_X and CS_GMA for MIPs and NIPs, respectively, where X was the percentage of 2,4-D template used for the formation of imprinted cavities. Briefly, 300 mg of CS:GMA was dissolved in 9 mL of 2% *v*/*v* acetic solution and stirred for 24 h. An amount of 1.7 g of ITA was added to the solution and left to stir until a homogeneous solution was obtained. Then 3 mL of DMSO containing 1.7 g of DMI and a different amount (50–200 mg) of the 2,4-D template was added to the aqueous solution. This latter was left under stirring for 24 h. After this time, 0.5 mL of ammonium persulfate (30 mg/mL) was used as the initiator (ca. 5% of the CS:GMA amount). The CS_GMA/ITA/DMI weight composition was equal to 8/46/46 while the weight percentage of template was in the range 1.3–5%. The mixture was centrifuged for 10 min at 6000 rpm to remove bubbles and then placed into a Teflon plate (about 7 cm in diameter). The polymerization was performed in an oven at 60 °C for 24 h. The same procedure was used to prepare NIPs (CS_GMA hydrogels without template). In the case of MIPs, after the polymerization phase, the template and unreacted monomers were removed with a series of washing phases using acetone, and ethanol. Then, to favour a complete extraction of 2,4-D pesticide, further washings first in methanol acidified with acetic acid (90/10) and then in pure methanol were carried out. The washing solutions were monitored by UV spectroscopy and HPLC until no trace of pesticide was detected in the washing solvents.

CS:ITA was used for the preparation of non-imprinted (CS_ITA) and imprinted (CS_ITA_X) hydrogels by adding to 10 g of a CS:ITA solution, 0.35 g of GMA and 3 mL of DMSO containing 0.172 g of DMI and different amounts of 2,4-D as a template. CS:ITA/GMA/DMI weight composition was equal to 72/18/10. The percentage of 2,4-D used for the imprinting process was in the range 2.5–9.5 wt(%). To start polymerization, 0.460 mL of an ammonium persulfate solution (37.5 mg/mL), corresponding to ca. 5% of the CS amount, was used as an initiator. The mixture was then centrifuged for 10 min at 6000 rpm to remove bubbles and placed into a Teflon plate (about 7 cm in diameter). The polymerization was performed in an oven at 60 °C for 24 h. CS_ITA (NIP) was prepared by following the same procedure but without the 2,4 D pesticide. Then, the same washing steps used for the CS_GMA systems were employed for CS_ITA samples.

#### Determination of Acrylation Degree of CS Functionalized with GMA

To determine the CS functionalization degree, ^1^H-NMR spectra of each modified chitosan sample were recorded with a Bruker Avance III HD-400AVIII spectrometer at 60 °C. Specifically, 10 mg of each sample was dissolved in 0.7 mL of deuterated acetic acid/water solution (10/90 *v*/*v*) and analyzed. The degree of acrylation (DA%) was determined by using the following equation [48]:$$\mathrm{DA}\;\%=\left[\frac{\left(A\delta \;5.9+A\delta \;6.3\right)/2\;}{A\delta \;3.03}\right]*100$$
where the area (*A*) of the signals at 5.9 and 6.3 ppm was attributed to the two hydrogen atoms of the vinyl group of GMA, while the area at 3.03 ppm was assigned to the H-2 glucosamine (GlcN) proton.

### 3.3. Characterization of the Prepared MIPs and NIP Systems

#### 3.3.1. Fourier Transform Infrared Spectroscopy

FT-IR spectra were acquired in attenuated total reflection (ATR) with a resolution of 4 cm^−1^ and coadding 200 scans by a Nicolet 6700 (Thermo Fisher Scientific, Waltham, MA, USA) equipped with a Golden Gate single reflection diamond.

#### 3.3.2. Scanning Electron Spectroscopy

The surface morphology of the prepared MIPs and NIPs was investigated by field emission scanning electron microscopy (FESEM, AURIGA Carl Zeiss AG, Oberkochen, Germany). For analysis, the membranes were fractured, chromium sputtered, and observed. Microanalysis was carried out with EDX (energy dispersive X-ray spectroscopy, Bruker Quantax, Berlin. Germany). EDX measurements were carried out both on the surface and in the bulk of the samples. The Cl _at.%_/C_at.%_ ratio was determined by the ratio of the atomic % of chlorine with respect to the atomic % of carbon. The resulting values were expressed in percentages.

#### 3.3.3. Thermogravimetric Analysis

The thermogravimetric analysis (TGA) was carried out by employing a Mettler TG 50 thermobalance (Mettler Toledo, Columbus, OH, USA). The sample analysis was performed under N_2_ flow, in the temperature range of 25–500 °C, by using a heating rate of 10 °C/min. Two scans were carried out on all of the samples.

#### 3.3.4. Water Uptake

To determine water uptake, weighted membranes were immersed in water (pH = 5) at room temperature. At different times, samples were taken, lightly dabbed on filter paper, and weighted again. The water uptake (W%) percentage was defined as follows:$$\mathrm{Swelling}\;\left(\%\right)=\frac{\mathrm{Wt}-W0}{W0}\times 100$$
where W0 and Wt are the weights of the samples at the initial and established time, respectively. Three different samples were tested, and the results are reported as average values. W (%) indicated the maximum swelling value obtained for each sample.

### 3.4. Study of Pesticide Adsorption

To test the adsorption capacity of the prepared MIPs and NIPs systems, about 20 mg of CS_GMA and CS_ITA series were put in contact with 2.2 mL of 2,4 D pesticide solution (10–100 ppm, pH = 5). A stock solution was prepared by dissolving a weighed amount of the pesticide in an appropriate methanol volume. Then pesticide solution was diluted with water at the chosen concentration. The pH of the pesticide solution was adjusted to 5 by adding HCl or NaOH. Quantitative analysis of 2,4-D adsorption was performed by a high-performance liquid chromatographer (Kontron Instrument HPLC PUMP 422) coupled with a UV–Vis detector (Jasco UV-2070, Easton, MD, USA). The system was equipped with a reversed-phase column C-18 (250 × 4.6 mm, particle size 5 µm). Injection volume and mobile phase flow were set at 20 µL and 1 mL/min, respectively. An isocratic mobile phase containing H_2_O/Acetonitrile 25:75 (acidified at pH 3.3 with acetic acid) was used, and the detector was set at a wavelength of 230 nm. A calibration curve was made to allow quantification. The 2,4-D pesticide adsorption was followed, and the quantities of 2,4-D absorbed at each time per g of sorbent phase were calculated (*q_t_*). While q_e_ indicated the quantities absorbed at plateau. Adsorption efficiency percentage (AE%) was expressed as follows:$$\mathrm{AE}\;\left(\%\right)=\left(\frac{Amount\;of\;desorbed\;pesticide\;\left(\mathrm{mg}\right)}{Initial\;pesticide\;amount\;\left(\mathrm{mg}\right)}\right)\times 100$$

Two desorption stages were performed by using 2 mL of methanol for one night and for two days, respectively. The quantity of desorbed pesticide was calculated for each fraction and the desorption efficiency (DE) percentages were calculated as follows:$$\mathrm{DE}\;\left(\%\right)=\left(\frac{Amount\;of\;desorbed\;pesticide\;\left(\mathrm{mg}\right)}{Amount\;of\;absorbed\;pesticide\;\left(\mathrm{mg}\right)}\right)\times 100$$

To examine the kinetic mechanism, pseudo-first-order and pseudo-second-order models were used. The following equations were used:$$\begin{array}{c} \left(I\right)\;\;\ln\left(q_{e}-q_{t}\right)=\ln q_{e}-K_{1}t\\ \left(\mathrm{II}\right)\;\;\frac{t}{q_{t}}=\frac{1}{K_{2}q_{e}^{2}}+\frac{1}{q_{e}} \end{array}$$
where *q_e_* = adsorption capacity at equilibrium (mg/g); *q_t_* = adsorption capacity at time t (mg/g); *t* = time (min); *k*_1_ (1/min) and *k*_2_ (g·mg^−1^·min^−1^) are the pseudo-first-order and pseudo-second-order rate constants, respectively.

### 3.5. Reusability of Systems

The reusability of the MIPs and NIPs systems was determined by using a pesticide solution (2.2 mL) at 100 ppm concentration for 3 repeated cycles. Briefly, the solution was put in contact with the sample (20 mg) for 120 min until the maximum adsorption was reached. Then, at the end of each assay, the sample was washed with methanol two times, dried and a new pesticide solution was added.

## 4. Conclusions

Molecularly imprinted hydrogels based on two types of acrylate chitosan were prepared for possible environmental applications. 2,4-D was used as a template in different amounts to test the effect of this parameter on the performance of the systems. In general, the structure of the samples was compact and not strongly influenced by the imprinting process. Only CS_GMA_5 had a slightly less dense network. The presence of residual pesticide entrapped in the matrix, revealed for the imprinted systems by EDX microanalysis, determined an increase in the thermal stability of the CS_GMA_X samples improving the interactions among CS chains. Whereas, for CS_ITA_X the residual pesticide acted as a disturbing agent reducing the T_d_. The swelling experiments evidenced a more permeable network for the CS_GMA-based systems. For these latter samples, a decrease in swelling was also observed when the residual pesticide entrapped into the matrices increased. Pesticide adsorption experiments, carried out on the prepared MIPs and NIPs systems, evidenced a correlation between the swelling ability and the pesticide uptake for the CS_GMA and CS_GMA_1.3 samples. On the contrary, a fairly good pesticide adsorption for the CS_GMA_5 sample if compared with the CS_GMA_2.5 system was found, despite its low swelling capacity. The result was justified by an increase in the number of cavities in the system, due to the high amount of the template used. The kinetic studies evidenced the adsorption process obeyed the pseudo-second-order kinetic model. CS_GMA_5 also showed a good possibility to be used repeatedly. Moreover, by reducing the pesticide concentration, CS_GMA_5 had the best desorption efficiency (90–100%). As for the CS_ITA-based systems, the best result was obtained for CS_ITA_1.3. In this case, probably the low amount of pesticide used for the MIP preparation could justify the increase in the adsorption efficiency. However, even if the CS_ITA_X systems showed lower performance than the CS_GMA-based system, they can be still considered promising because they are eco-friendly and guarantee a faster preparation procedure. These results suggest that a suitable strategy to obtain MIPs for the removal of contaminants is to develop sorbent materials with specific cavities starting with more environmentally friendly components such as modified chitosan. The performance of CS_GMA_5 has demonstrated it as a possible candidate for 2,4-D removal.

## Figures and Tables

**Figure 1 ijms-23-13192-f001:**
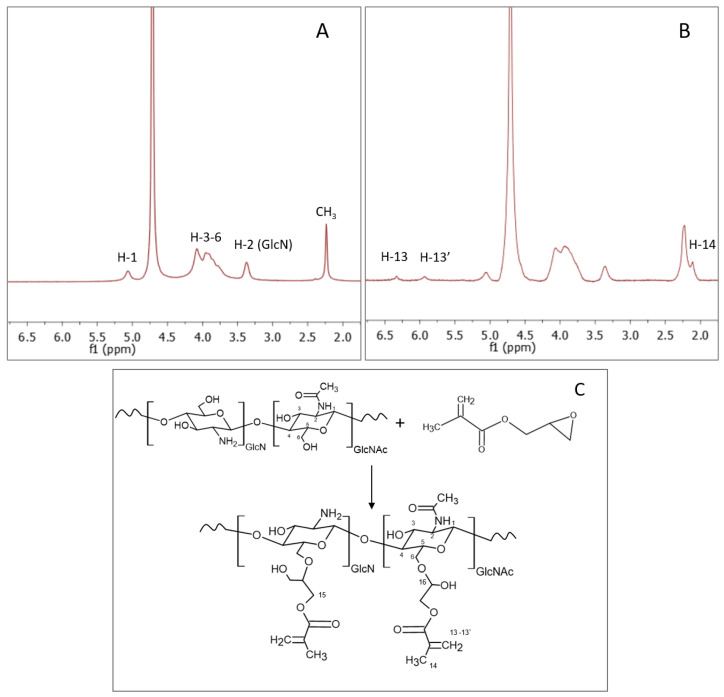
^1^H-NMR spectra of (**A**) pristine CS and (**B**) CS:GMA, and (**C**) scheme of CS modification with GMA.

**Figure 2 ijms-23-13192-f002:**
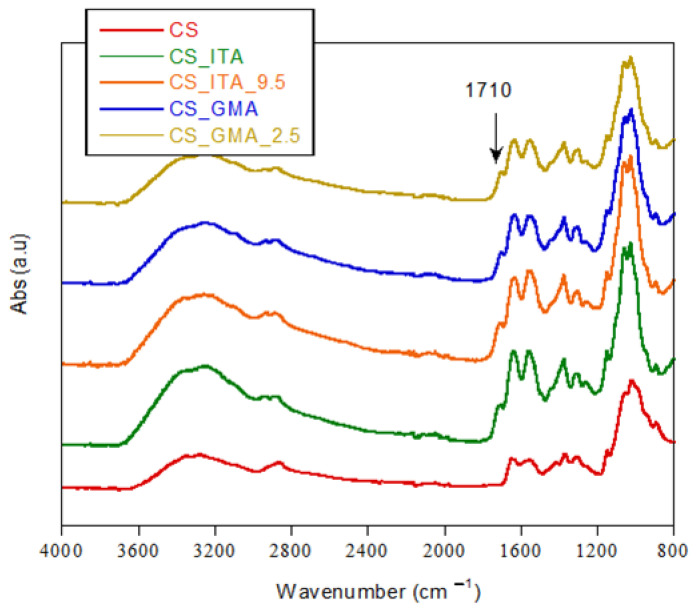
FT-IR spectra of some CS_GMA_X and CS_ITA_X MIPs and NIPs.

**Figure 3 ijms-23-13192-f003:**
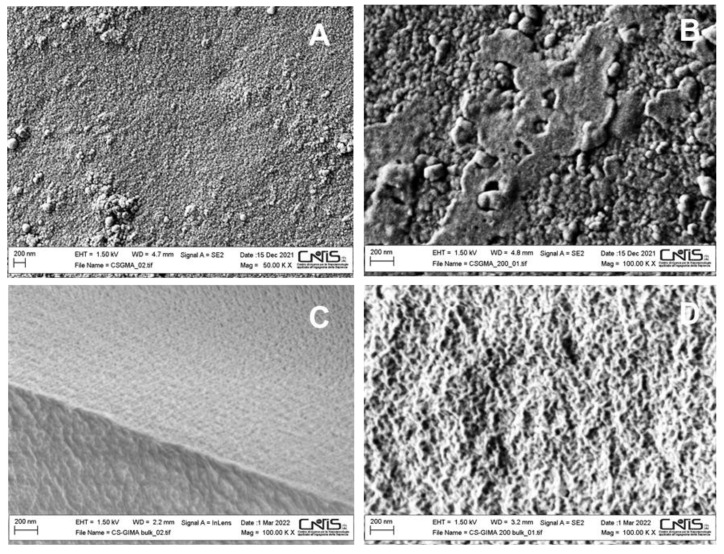
SEM images of the surface (**A**) and bulk (**C**) of CS_GMA, and surface (**B**) and bulk (**D**) of CS_GMA_5.

**Figure 4 ijms-23-13192-f004:**
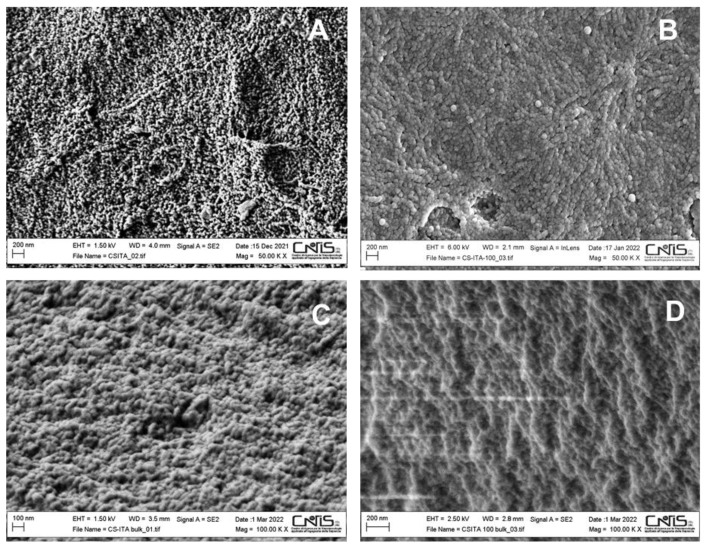
SEM images of the surface (**A**) and bulk (**C**) of CS_ITA, and surface (**B**) and bulk (**D**) of CS_ITA_5.

**Figure 5 ijms-23-13192-f005:**
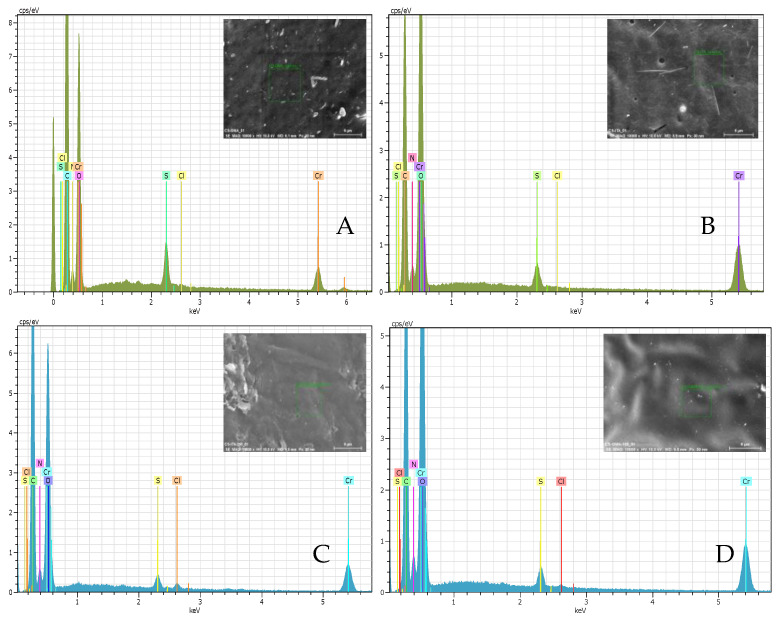
EDX spectra of (**A**) CS_GMA, (**B**) CS_ITA, (**C**) CS_ITA_9.5 and (**D**) CS_GMA_2.5.

**Figure 6 ijms-23-13192-f006:**
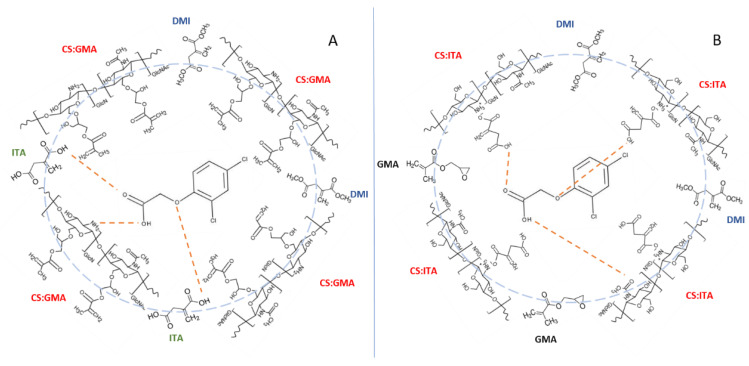
Interactions among the pesticide and MIP components for CS_GMA (**A**) and CS_ITA (**B**) systems.

**Figure 7 ijms-23-13192-f007:**
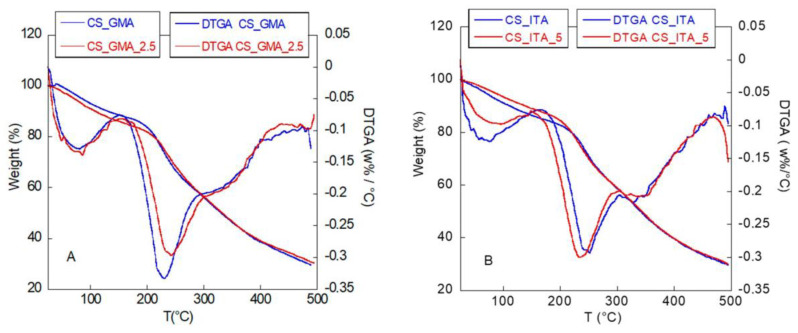
TGA and DTGA thermograms of CS_GMA (**A**) and CS_ITA (**B**) MIPs and NIPs.

**Figure 8 ijms-23-13192-f008:**
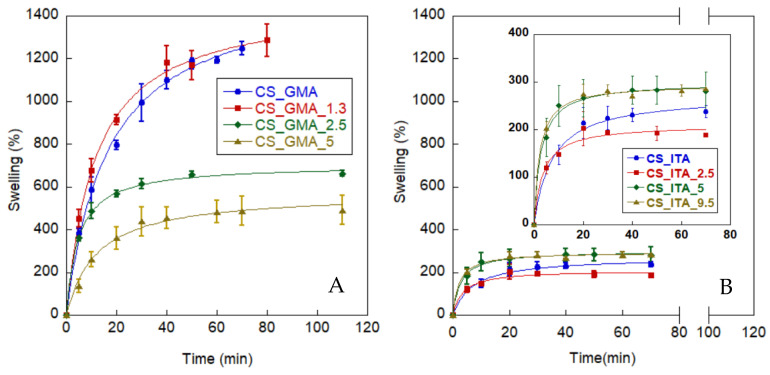
Water uptake kinetics of (**A**) CS_GMA MIPs and NIPs; (**B**) CS_ITA MIPs and NIPs.

**Figure 9 ijms-23-13192-f009:**
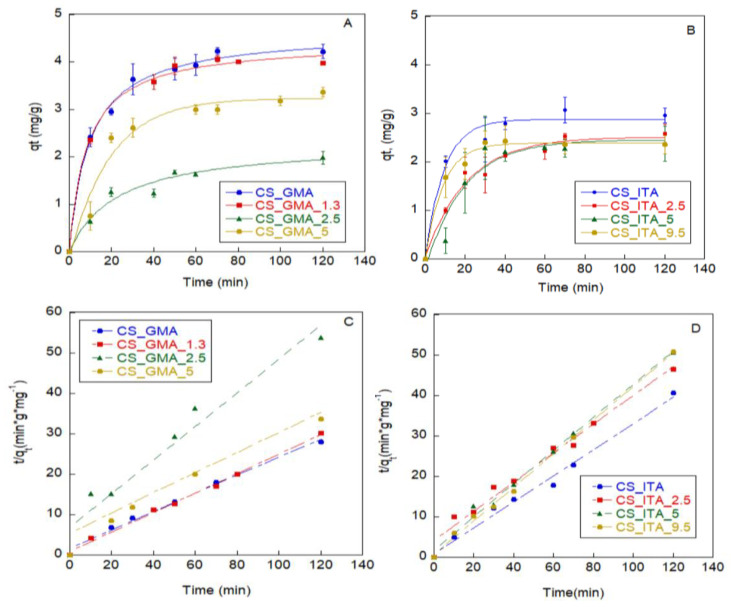
Adsorption kinetics of 2,4-D on CS_GMA (**A**) and CS_ITA (**B**) MIPs and NIPs; (**C**) pseudo-second-order model for CS_GMA and CS_GMA_X samples; (**D**) pseudo-second-order model for CS_ITA and CS_ITA_X samples.

**Figure 10 ijms-23-13192-f010:**
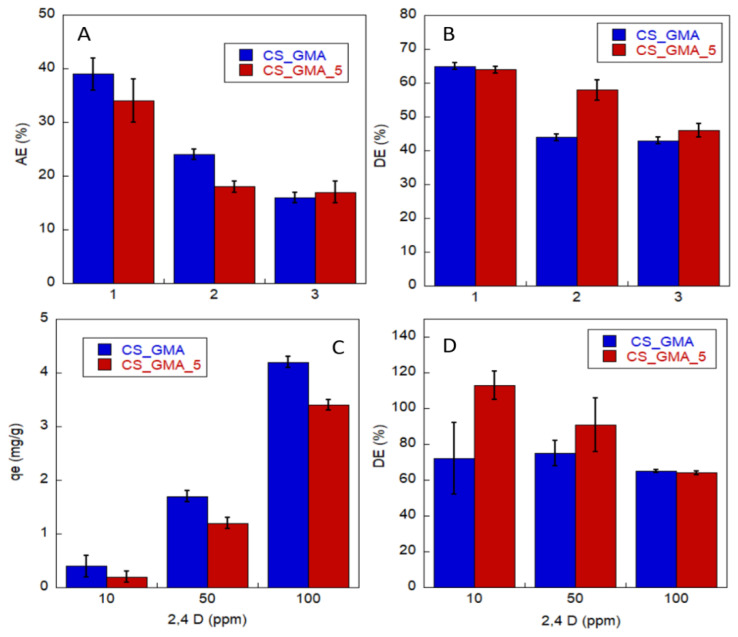
(**A**) Adsorption efficiency, AE (%) and (**B**) desorption efficiency DE (%) of CS_GMA and CS_GMA_5 samples for 4 consecutive cycles of sorption/desorption experiments carried out with a 2,4-D concentration equal to 100 ppm; (**C**) amount of 2,4-D absorbed per gram of materials and (**D**) desorption efficiency, DE (%) for CS_GMA and CS_GMA_5 by varying the pesticide concentration.

**Table 1 ijms-23-13192-t001:** Acronyms of CS_GMA and CS_ITA MIPs and NIPs.

Type of Modified CS	Imprinting	Template Amount wt (%)	Acronyms
CS:GMA	no	-	CS_GMA
	yes	1.3	CS_GMA_1.3
	yes	2.5	CS_GMA_2.5
	yes	5	CS_GMA_5
CS:ITA	no	-	CS_ITA
	yes	2.5	CS_ITA_2.5
	yes	5	CS_ITA_5
	yes	9.5	CS_ITA_9.5

**Table 2 ijms-23-13192-t002:** Summary of CS_GMA and CS_ITA properties. Cl _at.%_/C_at.%_ (%) ratio as revealed by EDX analysis, T_d_ is the temperature at which the maximum weight loss rate occurred. W (%) indicates the maximum swelling.

Sample		Physical Characterization		
	Cl _at.%_/C_at.%_ (%) Surface	Cl _at.%_/C_at.%_ (%) Bulk	Td (°C)	W (%)
CS_GMA	-	-	222 ± 3	1250 ± 30
CS_GMA_1.3	-	-	227 ± 2	1290 ± 70
CS_GMA_2.5	0.2 ± 0.1	-	228 ± 2	660 ± 20
CS_GMA_5	0.2 ± 0.1	-	230 ± 3	490 ± 70
CS_ITA	-	-	243 ± 1	238 ± 12
CS_ITA_2.5	0.2 ± 0.1	0.50 ± 0.05	229 ± 4	188 ± 10
CS_ITA_5	0.30 ± 0.03	-	230 ± 1	280 ± 40
CS_ITA_9.5	0.60 ± 0.04	0.70 ± 0.04	227 ± 2	286 ± 10

**Table 3 ijms-23-13192-t003:** Maximum amount of absorbed 2,4 D per gram of sorbent phase (q_e_); absorption efficiency (AE) and desorption efficiency (DE); rate constant of pseudo-second-order adsorption (K_2_); theoretical absorption capacity from pseudo-second-order model (q_e,cal_); correlation coefficient (R^2^).

Sample	q_e,exp_ (mg/g)	AE (%)	DE (%)	K_2_ (g·mg^−1^·min^−1^)	q_e,cal_ (mg/g)	R^2^
CS_GMA	4.2 ± 0.2	39 ± 3	65 ± 1	0.03422	4.4	0.9921
CS_GMA_1.3	4.0 ± 0.1	41 ± 1	45 ± 2	0.06972	4.1	0.9953
CS_GMA_2.5	2.2 ± 0.1	21 ± 1	95 ± 5	0.02139	2.4	0.9177
CS_GMA_5	3.4 ± 0.1	34 ± 4	64 ± 1	0.01205	3.8	0.9349
CS_ITA	3.0 ± 0.2	30 ± 1	66 ± 4	0.11756	3.1	0.9892
CS_ITA_2.5	2.6 ± 0.2	29 ± 1	54 ± 4	0.03117	2.8	0.9785
CS_ITA_5	2.4 ± 0.4	24 ± 2	72 ± 5	0.09963	2.4	0.9924
CS_ITA_9.5	2.3 ± 0.2	23 ± 1	66 ± 3	0.23976	2.4	0.9975

## Data Availability

The data necessary to reproduce the findings reported in this work can be obtained from the corresponding authors upon justified request.

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
