# Peer review of "Molecularly Imprinted Polymers Based on Chitosan for 2,4-Dichlorophenoxyacetic Acid Removal"

_ijms, 2022, doi:10.3390/ijms232113192_

Round 1

Reviewer 1 Report

This manuscript developed a low cost and eco-friendly materials for the removal of 2,4-Dichlorophenoxyacetic acid (2,4-D) from water. The molecularly imprinted polymers were prepared under optimized conditions starting from Chitosan (CS), chemically or ionically modified with glycidyl methacrylate (GMA) or itaconic acid (ITA), respectively. The influence of the template concentration on MIPs morphology, thermal behavior and swelling ability as well as on the 2,4-D removal capacity were analyzed. In addition, CS_GMA_5 had desorption efficiency of 90-100% when a low pesticide concentration was used. These findings exhibited that the presence of imprinted cavities could be useful in improving the performance of sorbent materials making CS_GMA_5 a possible candidate for 2,4-D removal. The preparation method of the adsorbent is interesting. However, the adsorption study of the adsorbent must be reviewed completely. Characterization of the adsorbent after adsorption, adsorption study and the data interpretation must be improved (discussion regarding the adsorption mechanism). The chart format in this manuscript should be amended according to this Journal. Overall, this paper is not suitable for publication in this form, which needs to be rewritten considering the recommendations mentioned below.

Some major concerns are shown as following:

1. Introduction: The introduction part should be restructured since it is considered too short and simple. Some related work should be reviewed in this part. Additionally, the reason why the authors selected this method is recommended to be added in the manuscript.

Regarding the removal of organic and inorganic pollutants with adsorbent, several critical papers should be added in the Introduction section, as this is important for readers to understand the recent works in this area.

For instance:

Journal of Cleaner Production, 2020, 120812. Polymers, 2019, 11(11), 1786. Int. J. Mol. Sci. 2022, 23(19), 11286

2. An illustration or a flow chart is recommended to further understand the reaction mechanism of the adsorbent.

3. The analysis for the FTIR pattern should rewritten since it is considered too simple. Additionally, the SEM examinations should be also rewritten for the same reason. All the figures in the manuscript should be redrawn using the software of Origin to ensure regular for the academic preciseness.

4. Why the authors not analyze the adsorbent after the adsorption?                        That would help a lot to determine the adsorption mechanisms involved in adsorption. If there is a possibility, please add this section to the manuscript.

5.Figure 7 and Figure 8: The most recent literature and some newly developed papers concerning the absorption should be cited to explain the phenomenon. How about the adsorption kinetic comparing your work to others?

6. The studies of adsorption kinetics are too simple.

7. Please check the entire manuscript for typos and unclear phrases by a native English writer.

Reviewer 2 Report

The manuscript presented a good concept of 2,4-Dichlorophenoxyacetic acid removal using CS-based MIP. However, some parts need to be improved before publication. The detailed comments are listed below.

Line 18: Give the full definition of the abbreviation of MIPs when it first appears, e.g., MIP (Molecularly Imprinted polymers). The same problem is shown in line 24: CS_GMA_5, line 138 (add the abbreviation FTIR in the bracket), in line 162 (full name first then abbreviation). Please check the same issues throughout the whole paper.

Line 97: Correct the number of Materials and Methods section 4.2 (should be 3.2).

Line 174: Fig.3c and 3d are not properly shown in the manuscript.

Line 179 and 199: Caption of Figure is confusing. Please change to A) B) C) D) making it more clear.

In my opinion, better to use word “adsorption” than “absorption”.

Line 318, the authors did not plot the kinetic studies of the adsorption experiment, therefore in the headline section 2.4, kinetics should not be used.

Line 386: In my opinion, as the removal efficiency was not so good, it should not claim that good performance.

Line 469: In 3.3.1. FTIR section, add the samples which were analyzed using FTIR.  Same problems in TGA part.

Adsorption mechanism should be discussed more in detail.

The manuscript presented a good concept of 2,4-Dichlorophenoxyacetic acid removal using CS-based MIP. However, some parts need to be improved before publication. The detailed comments are listed below.

Line 18: Give the full definition of the abbreviation of MIPs when it first appears, e.g., MIP (Molecularly Imprinted polymers). The same problem is shown in line 24: CS_GMA_5, line 138 (add the abbreviation FTIR in the bracket), in line 162 (full name first then abbreviation). Please check the same issues throughout the whole paper.

Line 97: Correct the number of Materials and Methods section 4.2 (should be 3.2).

Line 174: Fig.3c and 3d are not properly shown in the manuscript.

Line 179 and 199: Caption of Figure is confusing. Please change to A) B) C) D) making it more clear.

In my opinion, better to use word “adsorption” than “absorption”.

Line 318, the authors did not plot the kinetic studies of the adsorption experiment, therefore in the headline section 2.4, kinetics should not be used.

Line 386: In my opinion, as the removal efficiency was not so good, it should not claim that good performance.

Line 469: In 3.3.1. FTIR section, add the samples which were analyzed using FTIR.  Same problems in TGA part.

Adsorption mechanism should be discussed more in detail.

Round 2

Reviewer 1 Report

Comments for No. ijms-1953695 (International Journal of Molecular Sciences)

“Molecularly Imprinted Polymers Based on Chitosan for 2,4-Di-chlorophenoxyacetic acid Removal"
Corresponding Author: Antonella Piozzi
Recommendation - Accept.

Summary:

The authors have made substantial changes into the manuscript. Most of my comments were properly addressed by the authors and the paper was significantly improved, hence, I recommend it for publication.

Reviewer 2 Report

The manuscript can be accepted.